# Cystic Fibrosis Bone Disease: The Interplay between CFTR Dysfunction and Chronic Inflammation

**DOI:** 10.3390/biom13030425

**Published:** 2023-02-24

**Authors:** Óscar Fonseca, Maria Salomé Gomes, Maria Adelina Amorim, Ana Cordeiro Gomes

**Affiliations:** 1i3S–Instituto de Investigação e Inovação em Saúde, Universidade do Porto, 4200-135 Porto, Portugal; 2ICBAS–Instuto de Ciências Biomédicas de Abel Salazar, Universidade do Porto, 4030-313 Porto, Portugal; 3CHUSJ–Centro Hospital Universitário de São João, 4200-319 Porto, Portugal; 4IBMC–Instituto de Biologia Molecular e Celular, Universidade do Porto, 4200-135 Porto, Portugal

**Keywords:** cystic fibrosis related bone disease, osteoporosis, CFTR, bone

## Abstract

Cystic fibrosis is a monogenic disease with a multisystemic phenotype, ranging from predisposition to chronic lung infection and inflammation to reduced bone mass. The exact mechanisms unbalancing the maintenance of an optimal bone mass in cystic fibrosis patients remain unknown. Multiple factors may contribute to severe bone mass reduction that, in turn, have devastating consequences in the patients’ quality of life and longevity. Here, we will review the existing evidence linking the CFTR dysfunction and cell-intrinsic bone defects. Additionally, we will also address how the proinflammatory environment due to CFTR dysfunction in immune cells and chronic infection impairs the maintenance of an adequate bone mass in CF patients.

## 1. Introduction

Cystic fibrosis is a monogenetic autosomal recessive disorder with systemic and heterogeneous involvement [1]. The disease was first identified in 1938 [2]; then, the median life expectancy was a few months [3]. However, the implementation of newborn screening programs, the creation of reference centers to follow and treat cystic fibrosis patients, the elaboration of protocols to treat and eradicate chronic infection, and the improvement in the nutritional support, amongst other factors, increased the lifespan of cystic fibrosis patients to ~50 years old [1]. The improved longevity of cystic fibrosis patients creates new challenges, namely the understanding and treatment of several comorbidities such as cystic fibrosis bone disease [4].

Cystic fibrosis is most common in Northern Europe descendants but has been diagnosed worldwide [5], with increasing incidence in Asia and Africa [6,7]. According to the most recent patients registry reports, there are 52,246 patients in Europe and 31,411 in North America [8,9]. The disease is caused by mutations in the cystic fibrosis transmembrane conductance receptor (CFTR) [1,10,11], an ATP-binding cassette that also functions as a chloride and bicarbonate channel [12,13]. To date, over 2000 CFTR variations have been identified, but most of them are either very rare or have uncertain clinical significance [14,15]. The most frequent disease-causing mutation in European cystic fibrosis patients is the Phe508del (F508del). Individuals of other ethnicities are less likely to have this mutation, which has important therapeutic implications [16]. Mutations in CFTR with clinical phenotypes have a wide range of functional consequences and conceptually have been categorized in six classes (Figure 1). Whilst the severity of the clinical presentation correlates with the severity of the functional defect in CFTR (ranging from folding errors to complete lack of CFTR), several mutations associate with several functional deficiencies, exhibiting features of more than one mutation class [1,15]. Besides the particular CFTR mutation, the presence of different modifier genes and complex alleles in the same individual may also account for the differences in the disease severity and clinical manifestations [10].

In the lung and upper airways, dysfunction of CFTR leads to the formation of thick mucopurulent secretions and impaired mucociliary clearance, predisposing individuals to the establishment of chronic lung infections, mostly by *Staphylococcus aureus* and *Pseudomonas aeruginosa* [17]. The vicious cycle of infection and inflammation causes structural lung disease, such as the development of bronchiectasis, and in the latter and more severe stages of disease, respiratory failure [18]. The implications of chronic inflammation/infection in bone homeostasis will be discussed in Section 5.1.

Besides the effects on respiratory function, cystic fibrosis presents with endocrine alterations such as exocrine pancreatic insufficiency and cystic fibrosis-related diabetes. In pancreatic ducts, CFTR secretes chloride and bicarbonate, alkalinizing the ductal fluid, neutralizing peptic acid and optimizing pH for enzymatic digestion [19]. In the absence of CFTR function, ductal obstruction and epithelial damage, inflammation, fibrosis and fatty infiltration lead to an increase in pancreatic destruction [20]. In the most severe CFTR mutations, namely class I–IV and VI, these alterations begin in the uterus. Thus, pancreatic exocrine disease is a condition present at birth in almost all cystic fibrosis newborns. The small percentage of babies with cystic fibrosis that are pancreatic sufficient transition to insufficiency over time [20]. Pancreatic exocrine insufficiency is associated with malabsorption and liposoluble vitamin deficiency. Thus, lifelong pancreatic enzyme replacement and vitamins A, D, E and K supplementations are implemented in cystic fibrosis individuals with pancreatic exocrine disease [20]. The deficiency in vitamin D and its relationship with low bone mass is discussed below. Overtime, the pancreatic destruction leads to the loss of islet cells and the development of insulin deficiency and glucose intolerance. This condition is referred to as cystic fibrosis-related diabetes, and shares features with type 1 and type 2 diabetes mellitus [21]). The development of cystic fibrosis-related diabetes and insufficient insulin production might be relevant for reduced bone mass and will be discussed in Section 5.2.

The sequencing of CFTR in 1989 [22] was a turning point for the understanding of the disease pathophysiology, and for the search of a potential full-cure therapy. Gene therapy and heterologous or gene-corrected autologous stem cell transplantation have been proposed but with little clinical success [23,24]. Yet, a dramatic change in the therapeutical approach and in the hope of a full cure came with the development of CFTR modulators. These enclose several classes of small molecules that bind to CFTR, enhancing or even restoring the function of specific cystic fibrosis-causing mutants [25,26,27,28,29,30]. CFTR modulators were fast tracked for drug development and regulatory approval, with a timeframe of six years since the ivacaftor discovery and approval for treatment [26]. The European Medicines Agency (EMA) and the US Food and Drug Administration (FDA) have thus far licensed four products: ivacaftor (Kalydeco^®^, Boston, MA, USA) in monotherapy; the combination of ivacaftor with lumacaftor (Orkambi^®^, Boston, MA, USA) or with texacaftor (Symdeko^®^, Boston, MA, USA; Symkevi^®^, Dublin, Ireland); and the triple combination of ivacaftor with elexacaftor and tezacaftor (Trikafta^®^, Boston, MA, USA; Kaftrio^®^, Dublin, Ireland) [31]. These therapeutical options have improved the respiratory function and nutritional status of cystic fibrosis patients, overall enhancing quality of life without major safety concerns [32,33]. Of note, a recent study followed thirteen cystic fibrosis patients on ivacaftor/lumacaftor treatment and eight patients on ivacaftor/tezacaftor treatment and found that the circulating levels of pro-inflammatory cytokines such as TNFα decreased significantly with the treatment, indicating that combinations of CFTR modulators have potent anti-inflammatory proprieties [34]. Furthermore, some studies have demonstrated that ivacaftor enhanced the antimicrobial capacity of immune cells in cystic fibrosis patients treated with this CFTR modulator [35,36,37]). The understanding of how CFTR modulator therapy impacts extrapulmonary manifestations of cystic fibrosis will have the upmost importance for better understanding the pathophysiology of these manifestations and the search for directed and effective therapeutics against them [38]. The potential impact of CFTR modulators in cystic fibrosis-related bone disease will be discussed in Section 6.

## 2. Cystic Fibrosis-Related Bone Disease

Cystic fibrosis bone disease, accompanied by a severe bone mass reduction, is a common but not well understood complication in these patients, with devastating consequences to their quality of life and longevity. Bone loss in cystic fibrosis is multifactorial, and not very well understood. Indeed, the genetically determined CFTR malfunction coupled with the pro-inflammatory status, steroid therapy, amongst others factors, may all contribute to the dysregulation in the bone tissue (Figure 2).

The first report of reduced bone mass in patients with cystic fibrosis was in 1979 [39]. From then on, the prevalence of low bone mass in several cohorts of cystic fibrosis patients has been thoroughly studied, ranging from 20 to 35% [40,41,42,43,44,45]. Yet, the pathophysiology of the disease continues to be mostly unknown. Despite improved clinical management in terms of pulmonary function and vitamin D status, a significant and constant number of cystic fibrosis patients have reduced mineral density [46], emphasizing the importance of dedicated cystic fibrosis bone disease management. 

Physical activity, good exercise tolerance and nutritional status correlate with increased bone mineral density [47,48]. Early studies associated suboptimal vitamin D levels with cystic fibrosis bone disease [39,49], namely in individuals with end-stage lung disease [50]. However, the correction of vitamin D did not improve abnormalities in calcium, bone turnover markers [51], and bone mineral density [52]. Moreover, cystic fibrosis patients that underwenta lung transplant remained at risk of skeletal fragility [53], perhaps due to prolonged treatment with oral corticosteroids in the post-transplantation period. 

### Age and Bone Mass in Cystic Fibrosis

Cystic fibrosis bone disease has been mostly diagnosed in adult patients, but several studies indicate that bone defects appear early in life. Indeed, children with cystic fibrosis already have bone alterations [54,55,56,57,58] which increase with age [59]. The progressive bone loss [48] in cystic fibrosis patients correlates with reduced lung function [60,61,62], recurrent pulmonary exacerbations [63], inflammation [64] and reduced muscle strength [65]. Yet, these correlations and a reduced bone mineral density are not always identified in all cystic fibrosis populations [45], suggesting that other factors may also play a role in the development of bone loss in these patients [66]. Importantly, the development of osteoporosis in cystic fibrosis patients occurs with a deteriorating clinical status [67].

Cystic fibrosis bone disease predisposes individuals to low-impact fractures and skeletal abnormalities [68,69], decreasing the patients’ quality of life and even increasing the probability of a fatal outcome [70]. Moreover, cystic fibrosis bone disease is in some centers a counterindication for lung transplantation [71].

## 3. Cystic Fibrosis-Induced Alterations in Bone Architecture and Turnover

Bone is a specialized type of connective tissue that is responsible for the locomotion and protection of noble organs such as the brain, heart and lungs. Moreover, the skeletal system also functions as a mineral and hormone reservoir. Skeletogenesis starts during embryonic life through intramembranous or endochondral ossification (as reviewed in [72]). After birth and throughout life, bone mass and shape are regenerated due to bone modeling and remodeling. Throughout growth and aging, the bone shape adapts to variable mechanical demands through bone resorption and formation on opposing cortical and trabecular surfaces. This process refers to bone modeling, whereas the replacement of old or damaged bone by new bone in the same surface is called bone remodeling [73]. Bone remodeling or turnover is essential for an adequate maintenance of the bone mass throughout life and depends on a tight balance between the activity of bone-resorbing osteoclasts and bone-forming osteoblasts [74]. The imbalance in bone turnover is a natural consequence of aging, but it is often aggravated by several pathological conditions, namely those with an immunological etiology, such as HIV infection and rheumatoid arthritis [75]. Altered bone turnover may cause alterations in the bone architecture and strength, leading to bone microdamage that will be later translated into bone fragility [76]. Of note, reduced bone mineral density is not always the functional readout of an altered bone turnover. 

Cystic fibrosis bone disease is not restricted to a reduced bone mineral density. Structure and strength deficits in the bone of cystic fibrosis patients have been reported, namely at the cortical and trabecular levels [77], independently of the body mass index and size [78]. The bones of patients with cystic fibrosis with end-stage lung disease show reduced mass, altered microarchitecture, imbalanced remodeling and increased microdamage [79]. A histomorphometry analysis of the iliac bone of cystic fibrosis adults demonstrated an altered bone turnover with reduced mineral apposition, indicating reduced osteoblast activity [80]. Another study relying on histomorphometry analysis of postmortem vertebral bodies biopsy specimens demonstrated severe cortical and trabecular osteopenia with reduced osteoblasts and increased osteoclasts numbers [81]. This phenotype was also observed in mice deficient in CFTR [82]. Along these lines, an altered bone turnover with increased bone resorption and decreased new bone formation has been detected in patients with normal nutritional status and without acute pulmonary disease [83]. Indeed, reduced levels of osteocalcin (a biomarker of osteoblast activity) were found in cystic fibrosis patients with vertebral fractures [84]. In another study, besides reduced osteocalcin levels, an increased receptor activator of the nuclear factor kappa-Β ligand (RANKL)/Osteoprotegerin (OPG) ratio was found [54]. In bone metabolism, RANKL is a cytokine produced mostly by osteoblasts that binds to its receptor RANK expressed at the surface of osteoclasts and their precursors, thus inducing osteoclast formation and activity. Simultaneously, osteoblasts produce OPG, a decoy receptor for RANKL to control osteoclast differentiation and activity. An increased RANKL/OPG ratio indicates that an increased amount of RANKL is available for osteoclasts and consequently bone resorption is potentiated. Furthermore, the observations of reduced bone mineral density in young cystic fibrosis children with mild disease and normal nutritional status [85], as well as a reduction to half of the bone density in cystic fibrosis individuals compared to their healthy counterparts during childhood and puberty [86], point out that cystic fibrosis bone disease results from a primary defect in bone homeostasis. Furthermore, the dysregulation in bone homeostasis may be worsened by several periods of increased bone turnover and resorption during infective exacerbations [87].

## 4. CFTR Disfunction in Bone Cells and Their Progenitors

CFTR is expressed by several cells in the organism, namely by hematopoietic progenitors in the bone marrow and osteoclasts, as well as osteoblasts, osteocytes and chondrocytes and their progenitors [88,89,90,91]. Therefore, altered CFTR function may compromise osteoblast and osteoclast development and bone remodeling. The development of mouse models with *Cftr* deletion or with the F508del mutation have been fundamental for experimentally studying the role of CFTR in bone biology without confounders such as chronic lung infection and/or chronic malnutrition [82,92,93,94,95,96]. Furthermore, there is a correlation between the F508del mutation and low bone mass in mice [94,97], in agreement with previous observations in humans [92,98,99]. The absence of CFTR in newborn pigs led to an altered bone microstructure and chemical composition [100], and alterations in bone mineral density and architecture were found in *Cftr*-deficient mice and rats in the absence of other overt disease symptoms [93,101].

### 4.1. CFTR Dysfunction on Osteoblasts and Bone Formation

Osteoblasts are responsible for the secretion of the organic matrix composing the bone. These cells have a mesenchymal origin and mature into osteocytes when they become entrapped in the bone matrix. *Cftr* deletion hampers osteoblast differentiation [82,92,93,94]. Osteoblasts carrying the F508del mutation in CFTR exhibited a reduced expression of pro-osteoblastogenic factors such as the mothers against decapentaplegic homolog 2 (SMAD2), cyclooxygenase-2 (COX-2) as well as of OPG, but produced increased amounts of RANKL compared with osteoblasts with a fully functional CFTR [94,102,103]. These results indicate a possible defect in the activation of the Wnt/β-catenin signaling pathway that is fundamental for osteoblast maturation. Murine osteoblasts with F508del-CFTR demonstrated overactive nuclear factor kappa B (NF-kB) transcriptional activity, resulting in increased β-catenin phosphorylation and reduced β-catenin expression, as well as an altered expression of Wnt/β-catenin target genes [104]. Besides the Wnt/β-catenin signaling pathway, Wnt3a and parathyroid hormone-stimulated canonical Wnt signaling was also defective in mice lacking CFTR [96], as the lack of direct interaction between CFTR and dishevelled (Dsh) proteins prevents the stabilization of Dsh and its further interaction with Dpr1 [105,106]. Reduced Wnt/β-catenin signaling in F508del-CFTR osteoblasts was corrected via genetic or pharmacologic targeting of Keratin 8 [107], reinforcing that cystic fibrosis severity and associated comorbidities depend on the presence of different modifier genes and complex alleles [10]. Besides these molecular alterations, osteoblasts expressing F508del-CFTR secreted lower amounts of prostaglandin E2 (PGE_2_) and OPG [102,108]. Likewise, in lung cells, the absence of a functional CFTR chloride channel activity reduced the production of PGE_2_ [109]. PGE_2_ modulates Wnt/β-catenin signaling and induces osteoblast differentiation, maturation and activity [110,111,112]. Therefore, the reduced levels of PGE_2_ may account for the reduced bone formation. Besides the decreased OPG production, Tumor Necrosis Factor α (TNFα) stimulation and pharmacological inhibition of CFTR function together in osteoblasts in vitro decreased interleukin (IL) 8 secretion [108]. Higher circulating levels of IL-8 correlate with decreased femoral bone mineral density [113]. Additionally, TNFα and IL-17 further stimulate the production of RANKL via osteoblasts from cystic fibrosis patients [114]. RANKL is crucial for osteoclasts survival and differentiation [74].

Overall, CFTR dysfunction leads to decreased osteoblast formation and maturation (Figure 3).

### 4.2. CFTR Dysfunction and Osteoclastogenesis

Osteoclasts are giant multinucleated cells that originate from hematopoietic stem cells [74]. CFTR deficiency in mice was associated with increased numbers of osteoclasts and consequently to increased bone degradation [96]. Increased numbers of osteoclast progenitors were found in the peripheral blood of cystic fibrosis patients at the beginning of infective exacerbations and decreased after antibiotic treatment and resolution of the episode [115]. There are some conflicting results regarding the capacity for the osteoclast differentiation of peripheral mononuclear cells from cystic fibrosis patients. In one study, the formation of osteoclasts from circulating mononuclear cells was reduced [116], whereas in another study, peripheral blood mononuclear cells from cystic fibrosis individuals originated higher numbers of osteoclasts and bone resorption events when stimulated in vitro with osteoclast lineage-instructing cytokines [115]. These contradictory results could not be explained by the type of CFTR mutation present as both studies used cells from patients carrying at least one F508del allele, but perhaps differences in the patients age, inflammatory status and/or genetic background may explain these discrepancies. 

An alteration in CFTR function results in the activation of the NF-kB pathway and the secretion of several cytokines such as chemokine (C-C motif) ligand 2 (CCL2) upon bacterial infection or lipopolysaccharide stimulation [117,118,119,120,121,122]. Increased production of CCL2 induces osteoclast differentiation and bone resorption [123]. Indeed, osteoclast formation significantly correlated with the serum levels of TNFα, OPG, osteocalcin and *N*-telopeptide (NTx), and osteoclast activity significantly correlated with serum IL-6 and NTx [124]. Furthermore, monocytes from cystic fibrosis adults and pediatric patients express higher levels of macrophage colony-stimulating factor (M-CSF) receptors and show an increased propensity to differentiate into pro-inflammatory macrophages [125], suggesting that the enhanced osteoclastogenesis in these patients may be potentiated by recurrent inflammation or infection (Figure 4).

## 5. CFTR-Indirect Effects on Bone Health

### 5.1. Implications of Chronic Infection

Cystic fibrosis is associated with a hyperinflammatory state due to the conjugation of chronic lung infection, epithelial dysfunction and innate and adaptative immune dysregulation [126,127]. Alveolar macrophages and neutrophils accumulate in the lungs of cystic fibrosis patients [126]. Increased numbers of macrophages are found in the airways of cystic fibrosis newborns [128,129], and the majority of these cells are derived from circulating monocytes [130], suggesting an inflammatory bias in the lungs of these individuals. This basal inflammation may trigger alterations in bone homeostasis. Chronic infection of cystic fibrosis lungs perpetuates pro-inflammatory cytokine production, leading to elevated levels of TNFα, IL-6 and IL-1β and reduced levels of IL-10 [131,132,133]. Indeed, increased numbers of neutrophils and alveolar macrophages in the BAL fluid of *Cftr*-deficient mice at basal conditions impaired the control of *Pseudomonas aeruginosa* infection due to increased proinflammatory cytokine production such as TNFα, IL-1β and IL-6 [134]. Further supporting the role of the immune response in cystic fibrosis bone disease is the observation that the patients with a higher production of TNFα due to polymorphisms in *TNFA* and/or TNFα and TNFβ (LT-a) genes showed a decreased bone density [135]. TNFα induces osteoclast formation and bone resorption and impairs osteoblast formation [74,136,137]. 

The dysregulated pro-inflammatory response in the absence of CFTR is partially corrected via the transplantation of hematopoietic cells with normal CFTR function [138], suggesting that the pharmacological correction of CFTR functions may have several benefits in both lung health and other cystic fibrosis-associated diseases. 

### 5.2. Dysregulation of Glucose Homeostasis

Cystic fibrosis-related diabetes is a manifestation that usually appears around the second decade of life due to insulin deficiency [139]. Osteoblasts express insulin receptors and in response to insulin signaling, downregulate the expression of OPG and upregulate the production of osteocalcin [140,141]. Whereas the decrease in OPG increases the amount of RANKL available to bind to and activate osteoclasts and bone resorption, the production of osteocalcin has effects on whole-body glucose metabolism [142]. Undercarboxylated osteocalcin is an osteoblast-specific osteocalcin isoform that stimulates insulin secretion by pancreatic beta cells as well as increases insulin sensitivity in the liver, muscle and adipose tissue [143]. Besides osteocalcin, osteoblasts also produce the neuropeptide Y (NPY) that controls bone formation and also regulate glucose homeostasis. In the absence of the NPY receptor at an early stage of osteoblast differentiation, the bone mass increased but insulin secretion was reduced, leading to glucose intolerance [144]. 

In individuals without cystic fibrosis but with either type 1 or type 2 diabetes mellitus, the risk of fragility fractures is due to low bone turnover and macro and microarchitecture alterations [145]. Along these lines, a cohort of cystic fibrosis patients younger than eighteen years old who had impaired glucose tolerance or had already developed diabetes showed reduced bone mineral density and bone mineral content, indicating that poor glucose control together with other well-recognized factors such as reduced lung function, may impact bone homeostasis [146]. Furthermore, the patients with cystic fibrosis-related diabetes showed reduced bone turnover compared to patients without cystic fibrosis-related diabetes [147]. However, bone mineral density Z-scores were not different between the two groups of cystic fibrosis patients [147]. 

These observations highlight an intricate crosstalk between bone mass and glucose metabolism, suggesting that cystic fibrosis-related diabetes may be caused by factors other than pancreatic destruction. Additionally, the temporal relationship between cystic fibrosis-related diabetes and bone disease remains uncertain. Further studies will be required to address the relationship between low bone mass and impaired glucose homeostasis. 

## 6. CFTR Modulators and Their Potential Impact in Bone Health

Management of low bone mass in patients with cystic fibrosis relies on empirical non-pharmacological measures to optimize bone health and pharmacological measures such as bisphosphonates [148]. Bisphosphonates have been shown to increase bone mineral density in adults and children with this condition [55,149]. CFTR modulators pose a new therapeutic opportunity to these patients and have been associated with improvements in lung function [150]. On one hand, Interferon **γ**-mediated responses in cystic fibrosis monocytes were reduced after one week of treatment with ivacaftor [151]. On the other hand, whole-blood transcriptomic analysis revealed that innate and adaptative immune pathways persisted as overexpressed despite lumacaftor/ivacaftor treatment [152]. 

Even though it is still unclear whether the CFTR modulators ameliorate the disease-associated inflammation, there are some insightful observations regarding CFTR modulators and bone disease. Patients treated with ivacaftor showed an improvement in the microarchitecture of cortical bone [153] and in bone mineral density [154]. Additionally, TNFα and IL-17-induced RANKL production in osteoblast from cystic fibrosis patients was reverted via treatment with a CFTR modulator [114,154]. Oral administration of miglustat, a drug that improves F508del-CFTR function, improved bone mass and microarchitecture in the lumbar spine and femur of F508del-CFTR mice [95]. Pharmacological modulation of CFTR reduced the ratio of RANKL/OPG in osteoblasts with the delF508 mutation in CFTR [155].

## 7. Challenges in the Management of Bone Health in Cystic Fibrosis

Early detection of cystic fibrosis bone disease allows us to prevent its impact on the patients’ quality of life. For that purpose, the American Cystic Fibrosis Foundation and the European Cystic Fibrosis Society recommend the frequent monitorization of bone mineral density via a bone density scan (DXA) in all individuals with the disease [156]. Bone densitometry using DXA is a useful noninvasive method that allows for the assessment of the fracture risk in these patients [157] but is costly and exposes the patients to X radiation. Another caveat of DXA is the measurement of bone mineral density and the incapacity to detect alterations in the bone microarchitecture. Furthermore, the coverage of bone density screening via DXA is still low, as the median DXA screening rate of adults with cystic fibrosis in the US was only 66% [158]. The missed screening may result in an under diagnosis of cystic fibrosis bone disease, which may explain the decreased incidence of bone disease reported in the 2019 CFFPR report [8]. There is a need to identify better biomarkers and scores to stratify the patients in terms of the risk of bone disease and low impact fractures [159]. Perhaps the monitorization of the inflammatory status may predict the risk of low bone mass as the levels of C-reactive proteins negatively associated with the levels of procollagen type 1 *N*-terminal propeptide (P1NP) [147]. Thus, the monitorization of the inflammatory status in cystic fibrosis patients may be a tool to predict mineral bone mass.

## 8. Conclusions

Several factors contribute to alterations in the bone content and architecture in individuals with cystic fibrosis. The available data suggest that abnormal CFTR function causes an intrinsic defect in osteoblastogenesis, hampering new bone formation and the failure to reach an optimal bone mass peak and reduced bone turnover in cystic fibrosis patients. The chronic inflammatory status of these patients together with periods of infective exacerbations drives osteoclast formation and bone loss by further increasing the RANKL/OPG ratio. The impact in bone mass may also be potentiated by dysregulated glucose homeostasis and should be investigated in future studies.

The recently introduced CFTR modulators may be useful in decreasing the basal inflammatory status of people with cystic fibrosis, improving bone mass in these patients. Further studies are required to understand the impact of CFTR modulators in disease-associated inflammation and bone disease and to identify/characterize the relationship between chronic inflammation and reduced bone mass. Additionally, the development of better bone health screening biomarkers will be important for early detection and for acting on cystic fibrosis bone disease to prevent low-impact fractures and the consequent decay in the patients’ health and quality of life.

## Figures and Tables

**Figure 1 biomolecules-13-00425-f001:**
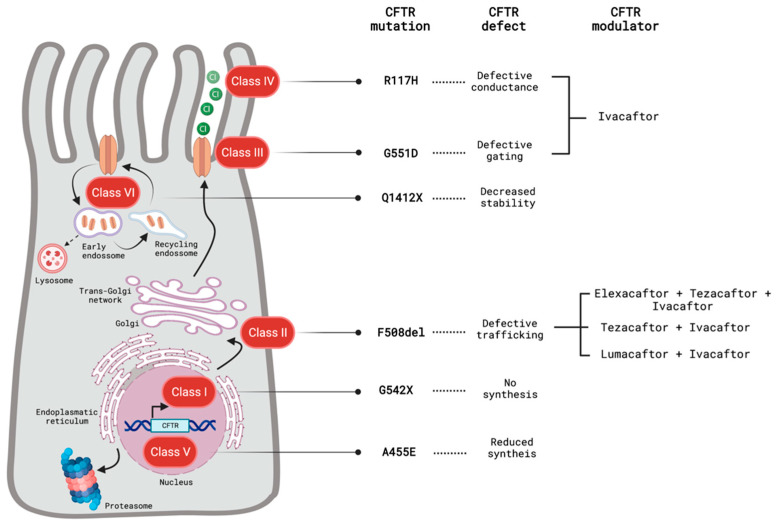
Classification and effects of CFTR mutations and approved CFTR function-restoring pharmacological interventions. Class I mutations result in the premature termination of CFTR transcription and in the absence of a functional CFTR. Class II mutations lead to the misfolding of CFTR protein and aberrant trafficking of the protein. Class III mutations are gating mutations causing ineffective chloride/bicarbonate transport. Mutations belonging to class IV decrease the conduction of ions through the channel. Class III and IV mutations lead to normal levels of CFTR expressed at the cell surface but a reduced CFTR function. Splicing and missense mutations belong to class V and cause reduced CFTR synthesis. Class VI mutations reduce CFTR stability of the cell membrane due to increased CFTR recycling. Thus far, the CFTR modulators in the clinic target class II, class III and IV mutations. Image created using Biorender.com (www.biorender.com; access on 22 November 2022).

**Figure 2 biomolecules-13-00425-f002:**
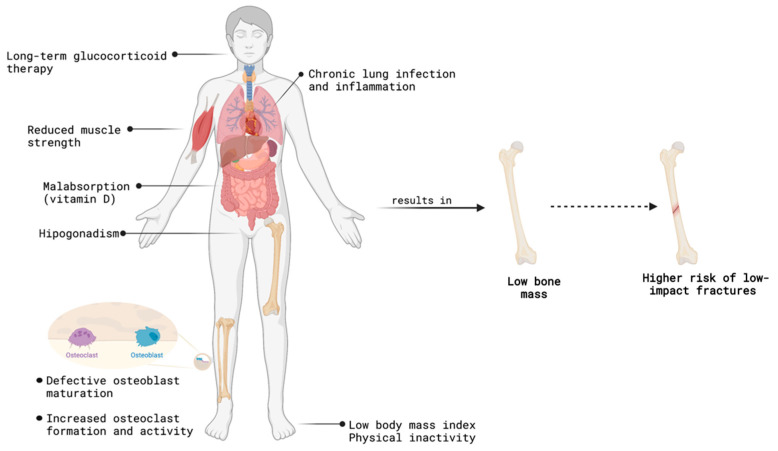
Cystic fibrosis bone disease is a multifactorial disease due to chronic organ involvement, the direct impact of CFTR dysfunction and long-term therapeutical approaches. Scheme was created using Biorender.com (www.biorender.com; access on 22 November 2022).

**Figure 3 biomolecules-13-00425-f003:**
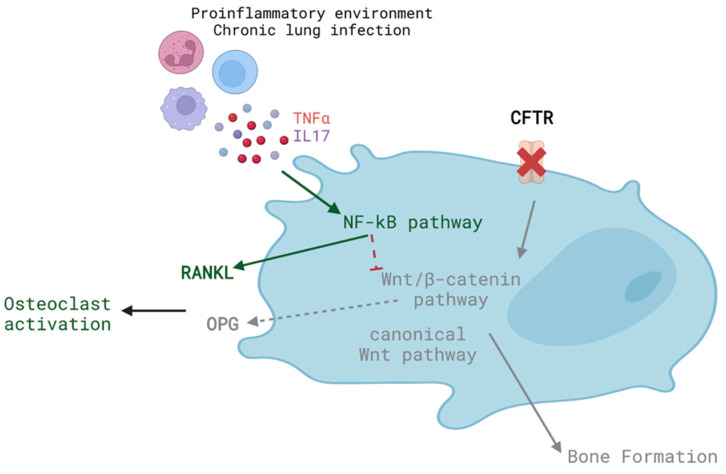
Defects in CFTR function hamper osteoblast maturation. Scheme was created using Biorender.com (www.biorender.com; access on 6 December 2022).

**Figure 4 biomolecules-13-00425-f004:**
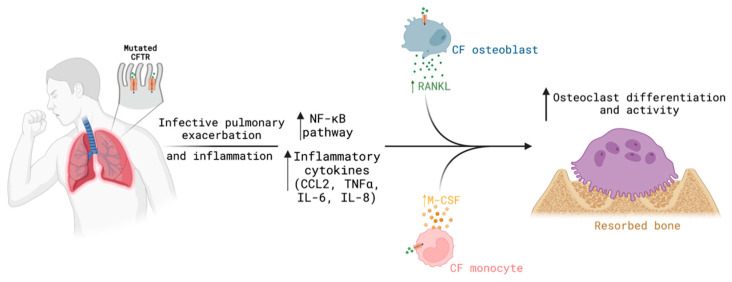
CFTR dysfunction and recurrent pro-inflammatory environment potentiates osteoclast formation and activity, contributing to bone loss and higher risk of fractures. Scheme was created using Biorender.com (www.biorender.com; access on 9 December 2022).

## Data Availability

Not applicable.

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
