# Peer review of "Cystic Fibrosis Bone Disease: The Interplay between CFTR Dysfunction and Chronic Inflammation"

_biomolecules, 2023, doi:10.3390/biom13030425_

Round 1

Reviewer 1 Report

The authors put together a comprehensive review of bone dysregulation in patients with CF. 

1. The title of the review can be better catered to the content of the review. The title implies that the review discusses the non-CFTR role in CF bone disease, however, the abstract and the content discuss the direct and indirect effects of CFTR at length and then cover a couple of other factors. 

1. For ease of the reader, the first paragraph of the review can be limited to CF and its lung disease manifestations. The systemic and other effects can be introduced in the second paragraph. 

2. The third paragraph (lines 58 to 100) needs to be more concise. Please condense to convey important points only for background and move it up to make it the first paragraph. 

3.  Although the sections contain a lot of relevant information, it needs to be organized better. Please restrict the content to related thoughts or ideas. It would make an easy read if you separate the direct effects of CFTR, the indirect effects of CFTR (CFTR-induced inflammation), and then other effects such as glucose levels .... more stringently. 

4. Also delineate as subsections the effects of age and treatment. 

Author Response

Reviewer 1

The authors put together a comprehensive review of bone dysregulation in patients with CF. 

  1. The title of the review can be better catered to the content of the review. The title implies that the review discusses the non-CFTR role in CF bone disease, however, the abstract and the content discuss the direct and indirect effects of CFTR at length and then cover a couple of other factors. 

We agree with the reviewer and changed the title of the review to “Cystic Fibrosis Bone Disease: the interplay between CFTR dysfunction and chronic inflammation”.

  1. For ease of the reader, the first paragraph of the review can be limited to CF and its lung disease manifestations. The systemic and other effects can be introduced in the second paragraph. 
  2. The third paragraph (lines 58 to 100) needs to be more concise. Please condense to convey important points only for background and move it up to make it the first paragraph. 

We thank the reviewer for his/her comments and suggestions. To address these two suggestions, we have reformulated the structure of the introduction (section 1, lines 24-114). The first paragraph (lines 25-33) briefly introduces cystic fibrosis. In the second paragraph (lines 34-60), the genetic basis of the disease are described to lay the ground on the direct effects of CFTR in bone cells (discussed in section 4, lines 451-549). In the third paragraph (lines 61-67), the pulmonary manifestations are introduced. This paragraph lays the ground for the discussion on the impact of chronic lung infection/inflammation in bone loss discussed in section 5.1 (lines 552-573). Then, in the fourth paragraph (lines 68-86), the extrapulmonary manifestations of cystic fibrosis are mentioned with emphasis on the pancreatic pathology. This information is important for the understanding of the tight regulation between bone and glucose metabolism which are further discussed in section 5.2 (lines 552-573). Finally, paragraph 5 (lines 87-114) introduces the novel treatment options for cystic fibrosis which are important concepts for section 6 (lines 640-659) that discuss the implications of CFTR modulators in cystic fibrosis bone disease.

  1. Although the sections contain a lot of relevant information, it needs to be organized better. Please restrict the content to related thoughts or ideas. It would make an easy read if you separate the direct effects of CFTR, the indirect effects of CFTR (CFTR-induced inflammation), and then other effects such as glucose levels .... more stringently. 

We thank the reviewer for the pertinent suggestion. In the revised section 4 (lines 451-549) of the manuscript, we address all the direct effects of CFTR-dysfunction both arms of bone homeostasis. For this purpose, two subsections were created. One to discuss the direct effect of CFTR dysfunction in osteoblasts (section 4.1, lines 464-507); the other section focus on the osteoclast intrinsic defects in the absence of a functional CFTR (section 4.2, lines 508-549). Then, the indirect effects of CFTR are discussed in section 5, which was divided in two subsections to discuss the implications of chronic infection to bone health (section 5.1, lines 552-573) and the dysregulation of glucose homeostasis (section 5.2, lines 552-573).

  1. Also delineate as subsections the effects of age and treatment. 

The effects of age in bone health are approached in section 2.1 (lines 308-321). Furthermore, the potential benefits of the treatment with CFTR modulators in the bone health of cystic fibrosis patients are discussed in section 6 (lines 640-659).

Reviewer 2 Report

The manuscript “Cystic Fibrosis Bone Disease: more than the dysregulation in the CFTR function” review how the CFTR dysfunction can impact the bone mass in CF patients. Authors identify chronic inflammatory status, glucose homeostasis dysregulation and decreased osteoblast/osteoclast maturation among the several factors that can alter the bone content and architecture.

The review is well written, scientifically sound and the data collected are meaningful in the field. Hence, it is suitable for publication after the following minor revisions:

-          When describing cohort studies, insert the NCT number to facilitate the identification of each clinical trial.

-          Authors should add a table summarizing the factors that affect bone mass in CF patients.

-          Lines 113-120 should be moved to paragraph 1, since they give general informations about CF that are more appropriate for the introduction section.

-          At the end of line 80-81, the authors mention the discovery of “several classes of small molecules that bind to CFTR, enhancing or even restoring the function of specific Cystic Fibrosis-causing mutations” but more recent and appropriate references are missing. Herein some examples: Molecules, 2021, 26(5), 1275 10.3390/molecules26051275; Journal of Medicinal Chemistry, 2020, 63(19), pp. 11169–11194; European Journal of Medicinal Chemistry, 2020, 204, 112631 https://doi.org/10.1016/j.ejmech.2020.112631; Science Advances, 2020, 6(8), eaay9669

-          The appropriate references are also missing at the end of line 310-311 and 328-329.

-          All acronyms should be spelt out (for ex. DXA).

-          English should be revised in some points (for ex. rephrase lines 317-318).

-          Few typo errors should be corrected. For ex. Cystic Fibrosis is written in capital letters, even in the middle of the sentence; in lines 212-218-223 CFTR is unnecessarily written in italics; in vitro and in vivo should be always written in italics.

-           

Author Response

Reviewer 2

The manuscript “Cystic Fibrosis Bone Disease: more than the dysregulation in the CFTR function” review how the CFTR dysfunction can impact the bone mass in CF patients. Authors identify chronic inflammatory status, glucose homeostasis dysregulation and decreased osteoblast/osteoclast maturation among the several factors that can alter the bone content and architecture.

The review is well written, scientifically sound and the data collected are meaningful in the field. Hence, it is suitable for publication after the following minor revisions:

-  When describing cohort studies, insert the NCT number to facilitate the identification of each clinical trial.

We thank the reviewer for the thoughtful suggestion. We agree with the reviewer and looked for the NCT number of all the cited references. Unfortunately, only 2 out of 159 references contained a NCT number. Those references were the following articles:  Bianchi et al 2013, Lancet Resp. Med. and Putman et al, 2021, J Clin Endocrinol Metab. This is most likely explained by the fact that most of the cited studies are not interventive but observational studies. Therefore, we decided to not add the NCT numbers in the manuscript.

- Authors should add a table summarizing the factors that affect bone mass in CF patients.

We thank the reviewer for the suggestion but instead of a table we added a figure (Figure 2, lines 290-293) that illustrates/ summarizes the factors that affect bone mass in CF patients.

-  Lines 113-120 should be moved to paragraph 1, since they give general informations about CF that are more appropriate for the introduction section.

We have reformulated the manuscript to include the reviewer suggestions. We have reformulated the structure of the introduction (section 1, lines 24-114). The first paragraph (lines 25-33) briefly introduces cystic fibrosis. In the second paragraph (lines 34-60), the genetic basis of the disease are described to lay the ground on the direct effects of CFTR in bone cells (discussed in section 4, lines 451-549). In the third paragraph (lines 61-67), the pulmonary manifestations are introduced. This paragraph lays the ground for the discussion on the impact of chronic lung infection/inflammation in bone loss discussed in section 5.1 (lines 552-573). Then, in the fourth paragraph (lines 68-86), the extrapulmonary manifestations of cystic fibrosis are mentioned with emphasis on the pancreatic pathology. This information is important for the understanding of the tight regulation between bone and glucose metabolism which are further discussed in section 5.2 (lines 552-573). Finally, paragraph 5 (lines 87-114) introduces the novel treatment options for cystic fibrosis which are important concepts for section 6 (lines 640-659) that discuss the implications of CFTR modulators in cystic fibrosis bone disease.

-  At the end of line 80-81, the authors mention the discovery of “several classes of small molecules that bind to CFTR, enhancing or even restoring the function of specific Cystic Fibrosis-causing mutations” but more recent and appropriate references are missing. Herein some examples: Molecules, 2021, 26(5), 1275 10.3390/molecules26051275; Journal of Medicinal Chemistry, 2020, 63(19), pp. 11169–11194; European Journal of Medicinal Chemistry, 2020, 204, 112631 https://doi.org/10.1016/j.ejmech.2020.112631; Science Advances, 2020, 6(8), eaay9669

We agree with the reviewer and apologize for missing these references, which have added the revised manuscript (references 27 to 30).

- The appropriate references are also missing at the end of line 310-311 and 328-329.

We apologize for the missing citations. We have corrected this mistake by adding reference 108 in line 491, and reference 147, lines 630-631.

- All acronyms should be spelt out (for ex. DXA).

We apologize for the mistake and have spelt out all acronyms throughout the manuscript.

- English should be revised in some points (for ex. rephrase lines 317-318).

We apologize for the mistake and have revised the text namely in lines 619-620 (former lines 317-318) as noted by the reviewer.

- Few typo errors should be corrected. For ex. Cystic Fibrosis is written in capital letters, even in the middle of the sentence; corrected in lines 212-218-223 CFTR is unnecessarily written in italics; corrected in vitro and in vivo should be always written in italics.

We apologize for the typos and have corrected them. Regarding the CFTR written in italics, in lines 456, 462 and 467, we are referring to the murine gene and therefore the correct nomenclature is used. 

Round 2

Reviewer 1 Report

The authors has reorganized the content and addressed the concerns mentioned satisfactory.

Author Response

We thank the reviewer for recognizing our efforts to address his/her concerns.

The reviewer requested moderate English changes but did not specify what are the required changes. Therefore, our manuscript has been revised by proficient English speakers to improve the English language and style.